# Microstructure and Phase Evolution of Ti-Al-C-Nb Composites Prepared by In Situ Selective Laser Forming

**DOI:** 10.3390/ma15124351

**Published:** 2022-06-20

**Authors:** Kai Zhang, Shurui Li, Zhilong Yan, Zhiwei Xiong, Desti Dorion Likoundayenda Bakoutas, Tingting Liu, Wenhe Liao

**Affiliations:** School of Mechanical Engineering, Nanjing University of Science and Technology, Nanjing 210094, China; zhangkai@njust.edu.cn (K.Z.); lishurui@njust.edu.cn (S.L.); yanzhilong2022@163.com (Z.Y.); zhiweixiong111@163.com (Z.X.); destidorion@gmail.com (D.D.L.B.)

**Keywords:** in situ selective laser forming, Ti_3_Al, TiC, reaction, porosity

## Abstract

In the present work, a novel Ti-Al-C-Nb composite was prepared using in situ selective laser forming (ISLF). The formation mechanism of the Ti-Al-C-Nb bulks, which were synthesized using elemental titanium, aluminum, and carbon (graphite) powders via ISLF techniques, was investigated. The results showed that the Ti_3_Al and TiC phases were the dominant synthesis products during the chemical reactions, and these occurred during the ISLF process. The size of the fine nanoscale crystal TiC grains could reach 157 nm at an energy level of 60 J/mm^3^. The porous structure of the ISLF specimens was disclosed, and an open porosity of 20–44% was determined via the scanning speed and the laser power. Both the high dynamic viscosity and the reactions of the raw powders led to the generation of a considerable number of pores, whereas the specimen processed using 45 W and 100 mm/s possessed the lowest degree of open porosity.

## 1. Introduction

Metal–ceramic composites integrate the advantages of each component, and they have potential application prospects in the fields of aerospace, industrial metallurgy, and machinery [1,2,3]. In the last few decades, Ti-Al-C composites, which contain the TiAl_3_ phase and TiC particles, have aroused great interest in the above fields, due to their excellent hardness, thermal stability, good wear resistance, and corrosion resistance [4,5]. Presently, the comprising parts of the Ti-Al-C composites can be prepared using many different methods, such as self-propagating high-temperature synthesis [6], aluminum melt reaction [7], and laser cladding deposition [8]. However, these methods are limited by their degree of reaction control difficulty, the complexity of the processes, and the mold-limiting complexity of forming, which hinders their practical applications.

Selective laser melting (SLM), as a powder material-based additive manufacturing technology, can achieve the rapid prototyping of complex structures with good forming accuracy [9]. Although many studies have been conducted on the SLM of metal materials, the work that is focused on metal composites containing ceramic phases is rarely mentioned. Chen et al. conducted variable laser process parameter experiments using the SLM process, and successfully prepared a Ti-Si-C composite material bulk [10]. They found that increasing the laser energy density can reduce the porosity of Ti-Si-C samples to further prepare functionally gradient materials. Ma et al. [11] used Ti, Al, and TiC mixed powders to synthesize TiC ceramic-reinforced TiAl-based composites through SLM technology. They found that in high-temperature environments, multilayer oxide films that form on the surfaces of TiC/TiAl samples have a certain high-temperature oxidation resistance (1.32 × 10^−5^ mg^6^ cm^−12^ h^−1^) [11]. Shao et al. studied the SLM forming process of the Al-TiO_2_-TiC-CNT powder system and found that the chemical reaction inside an SLM sample with a phase composition of TiAl_3_/TiC becomes more sufficient as the laser power increases [12]. Research on the Ti-Al-C system in the SLM process remains at the level of the preparation process and microscopic characterization, and few studies have focused on the internal defects of samples, especially regarding the generation of pores.

Although the TiAl phase, which is one of the multiple phases in the Ti-Al-C system, has excellent oxidation resistance and a high elastic modulus, the low high-temperature strength and toughness of TiAl reduces the mechanical properties of the composite material [13]. The element Nb has excellent ductility, low density, and high toughness, perfectly compensating for this deficiency [14,15]. Wei et al. found that (Nb,Ti)C and Nb3Al, which form in Nb-Ti-Al-C alloys, can be used as effective reinforcing phases to improve the mechanical properties of Nb-based alloys at high temperatures [16]. Therefore, adding a small amount of Nb to enhance the Ti-Al-C system is a significant attempt at improving the system.

In our previous research [17], we proposed a new forming process—that is, in situ selective laser forming (ISLF), which is defined based on SLM. The difference between ISLF and SLM is that the target-shaped portion is formed in situ in ISLF. In this work, we conducted ISLF tests with mixed Ti, Al, graphite, and Nb powders as raw materials, mainly studying the influence of process parameters on the phase composition and microstructure of the sample and analyzing the ISLF sample forming process. The chemical reactions between the phases and the evolutionary behaviors of TiC grains were studied to explore the reasons for the porous structure of the sample. The influence of Nb is not discussed in this paper, and we will analyze its influence law more specifically in another paper.

## 2. Materials and Methods

### 2.1. Powder Preparation

The raw powdered materials used in this work included Ti powder (99.99 wt% purity, +300 mesh), Al powder (99.95 % purity, average particle size is 50 μm), graphite (99.95 wt% purity, 750–850 mesh), and Nb powder (99.95 wt% purity, +325 mesh). All of the raw materials mentioned are commercially available. The morphologies and particle sizes of the powders are presented in Figure 1. The powders were weighed in a certain proportion (70.2 wt% Ti, 20 wt% Al, 8.8 wt% graphite, and 1 wt% Nb) and milled in a planetary mill (QM-3SP4, Nanjing, China) at 150 rpm for 8 h. The weight ratio of the aluminum balls to powder was 2:1. Absolute ethanol was added during milling to boost the mixing. The wet powder was dried in 120 °C for 5 h in a vacuum drying oven.

### 2.2. Process

The self-developed ISLF equipment (Nanjing University of Science and Technology, Nanjing, China) mainly consists of a YLR continuous fiber laser (λ = 1.06 μm) with a maximum output power of 500 W and a protective atmospheric system. Ar was used as the protective gas to avoid oxidation. The oxygen and water contents in the working chamber were controlled at 20–50 ppm during the ISLF process. The processing parameters were optimized as follows: laser power values of 35, 40, 45, and 50 W; scanning speeds of 100, 150, 200, 250, and 300 mm/s; a hatch space of 50 μm; and a layer thickness of 50 μm. The zigzag scanning strategy adopted in the ISLF process is shown in Figure 2a. The direction of the laser scanning path between two layers was rotated by 90°. The dimensions of the ISLF samples were 10 mm × 10 mm × 6 mm. The samples printed using different process parameters are shown in Figure 2b. It should be noted that the samples processed at 35 W and 300 mm/s warped during the initial printing, and the sample was broken after cutting due to the large internal stresses imposed, and so it was not characterized using SEM and XRD in this this paper. To better study the influence of laser energy on the experimental results, the concept of laser energy density (*η*) was established (Table 1):(1)$$\eta =\frac{P}{vhd}$$
where *η* represents the energy density (J/mm^3^), *P* represents the laser power (W), *v* represents the scanning speed (mm/s), *d* represents the hatch spacing (mm), and *h* represents the layer thickness (mm).

### 2.3. Characterization

The ISLF samples were removed from the substrate using wire electrical discharge machining, and polished with diamond sandpaper of different meshes from #400 to #7000. The overall porosity of the samples was measured using the Archimedes drainage method (based on ASTM C20-00) impregnated with absolute ethanol [18], and calculated using the following formula:(2)$$\rho =\frac{dry\;mass\cdot \rho_{ethanol}}{wet\;mass-suspended\;mass}$$
(3)$$Porosity_{overall}(\%)=(1-\frac{\rho }{\rho_{theorethical}})\cdot 100$$
where “dry mass” is the mass (in g) of the dry sample, “wet mass” is the mass of the sample after soaking in absolute ethanol for 2 h, “suspended mass” is the mass of the sample suspended in ethanol using a suspending system, and $\rho_{ethanal}$ = 0.79 g/cm^3^ is the density of ethanol. $Porosity_{overall}$ is the volume percentage of the overall porosity of the sample. The theoretical density of the powder, $\rho_{theorethical}$, was determined using the proportion of phases in the sample. All the data were obtained after three measurements were averaged.

X-ray diffraction (XRD, Bruker-AXS D8 Advance, Karlsruhe, Germany) using a Cu anode wavelength of 1.542 Å was adopted to measure the phase composition of the sample. Specifically, the respective current and voltage were 40 KA and 40 KV, and continuous scan mode was adopted, with a scanning step of 0.02° and a scan time per step of 0.2 s. The open-access databases COD and Profex 5.0 were used to analyze the XRD results via the Rietveld method [19]. Field emission scanning electron microscopy (SEM, FEI Quanta 250F, Brno, Czechia) was used to observe the microstructure of the sample, and a scanning electron microscope was equipped with an energy dispersive X-ray detector (EDS, Oxford Instruments, UK) for elemental distribution detection.

ImageJ was used to analyze the average area of the holes in the low-magnification SEM image of Figure 10. The calculation process of the area of the average pores was as follows: Extract the number of pixels in each low-magnification SEM image, and the number of pixels per unit length (mm). Then, extract all pores, analyze the proportion of the pores, and count the number of holes in each image. Thus, we defined the average area of the holes as *S*:(4)$$S=\frac{bc}{d^{2}}\cdot \frac{p}{n}$$
which is defined by pixels *b* and *c*, the number of pixels per unit length mm *d*, the porosity *p*, and the number of holes *n*.

ImageJ software was used to analyze low-magnification SEM images. A mixed solution of HF, HNO_3_, and H_2_O was used, with a volume ratio of 5:25:50, to etch the upper surface of the polished sample for 10 s to observe the microstructure.

## 3. Results

### 3.1. Phase Analysis and TiC Grain Morphology

Figure 3 shows the SEM, EDS, and XRD results of the mixed powder after sieving and drying. The milling process did not change the morphologies and sizes of the raw powders, and each powder existed independently and was uniformly dispersed, providing the basis for a sufficient reaction between the powders during the ISLF process. The XRD (Figure 3f) spectrum only contains the diffraction peaks of the three original powders of Ti, Al, and graphite, indicating no chemical reaction during the ball milling process and no new phases formed in the mixed powder. The diffraction peak of Nb did not appear in the XRD pattern, probably because the Nb content in the mixed powder was extremely low. The Nb phase was not detected by XRD.

Figure 4 shows the XRD patterns of the ISLF samples at different scanning speeds. The matching of the XRD data revealed that the diffraction peaks corresponding to the initial powder basically disappeared, and multiple new diffraction peaks appeared in the XRD patterns of the sample. The strong diffraction peak was a combination of Ti_3_Al (mp-1823) and TiC (cod-5910091), which is common among the XRD patterns of all the samples, indicating that the main phases of the ISLF samples consisted of Ti_3_Al and TiC. Meanwhile, the weak diffraction peaks of TiAl (mp-1953) and graphite (cod-900046) indicated that the secondary phase consisted of TiAl and raw material graphite. The low graphite diffraction peaks indicated a small amount of graphite in the sample. Table 2 shows the mole fraction and theoretical density of each phase in the ISLF samples after calculation. As can be seen from Table 2, Ti_3_Al, TiAl, and TiC were the main phases, with the amounts varying from sample to sample. There was no difference in quantity for the different process parameters. The greatest phase quantities were Ti_3_Al, then TiC. The quantity of TiC was approximately 30–40%.

The XRD results showed that the TiC content, which is one of the main phases in the ISLF sample, changed regularly with different process parameters. The microscopic morphology of the TiC grains was analyzed to better understand the reasons for the changes in TiC phase composition during the ISLF process. Figure 5 shows the microstructure of the ISLF samples after corrosion under different process parameters. The figure shows that the surfaces of the samples had mainly white particles. From Table 3, we can see that the atomic ratio of Ti and C was close to 1:1. It should be noted that the detection of C by EDS has certain inaccuracies. In addition, the white particle phase showed a dendrite growth mode [11], and from analysis, it was found that the white particles were mainly TiC grains. The TiC crystal grains showed different distributions of microscopic morphology with an increase in laser energy density.

When the laser energy density was 60 J/mm^3^, the input energy was low, the TiC crystal grains were elliptical, the grain size could reach 157–254 nm, and the elliptical grains were randomly distributed. The lengths of a few strip crystals were approximately 0.82–1.08 μm (Figure 5a,b). When the laser energy value was increased to 80 J/mm^3^, the elliptical TiC grains grew into uniformly distributed spherical nano-crystalline grains, and the grain size increased further, reaching 215–482 nm (Figure 5c,d). When *η* was 160 J/mm^3^, the TiC grains grew to micron-level strip crystals. The strip crystals were mainly formed from multiple small nanocrystals that were stacked in the axial direction, with an approximate length of 1.41 μm, and a relatively smaller dendritic arm spacing; the mean thickness of the dendritic arms was 0.19 μm (Figure 5e,f). When *η* increased further to 200 J/mm^3^, the long and narrow stripe crystals grew into coarse dendrites along the secondary crystal axis direction perpendicular to the primary crystal axis, and the dendrite length could reach 5.01–7.05 μm; the thickness of the dendritic arms increased to 0.34 μm (Figure 5g,h).

### 3.2. Microstructure

Figure 6 and Figure 7 are the SEM images of the surface morphologies of the ISLF samples. The figure shows that the surfaces of the ISLF samples were rough. Obvious spheroidization could be observed on the surfaces of the samples. At the same time, obvious cracks and pores were observed on the surfaces of the samples, as shown in Figure 6b,c.

Figure 6 is the SEM image of a variable laser power ISLF sample with a scanning speed of 150 mm/s. When the laser power was 35 W, the solid part of the sample surface consisted of cracked spheroidal solidified parts, and the spheroidization was most obvious (Figure 6a). When the laser power increased to 40 W and 45 W, the continuity of the solid part of the sample surface improved, and the number of irregular holes decreased (Figure 6b,c). When the laser power increased to 50 W, the surface state of the sample did not improve further (Figure 6d). Figure 7 is the SEM image of an ISLF sample with a laser power of 40 W and a variable scanning speed. When the scanning speed continued to increase from 100 mm/s to 300 mm/s, the spheroidization of the sample surface intensified, and the continuity of the solid part continued to decrease. When the scanning speed was 200 mm/s, the sample surface transitioned to a single solidified melt splicing state. The size of the quasi-spherical solidified melt was approximately 200 μm. Figure 8 presents the cross-sections of the ISLF samples under different laser powers. It was found that the samples exhibited porous structures. As the laser power increased, the number of holes and the number of connected holes decreased to a certain extent.

The open porosities of the ISLF samples were calculated using the Archimedes drainage method, using the measurement and phase content results. Figure 9 shows the overall porosities of all of the ISLF samples. The aperture ratios of the samples varied from 20% to 44%. The lowest opening rate of 20% corresponded to the process parameters of 50 W and 100 mm/s, which were the parameters with the highest input energies. When the scanning speeds were 100 mm/s and 300 mm/s, Porosity_overall_ decreased with an increase in laser power. It should be noted that this trend could not be seen in the cases of the 150 mm/s, 200 mm/s, and 250 mm/s scanning speed results. The mechanism of overall porosity for the ISLF samples will be discussed in Section 4.2.

Figure 10 is the low-magnification SEM image of the ISLF sample after grinding and polishing. All of the low-power SEM images are mainly composed of three areas: gray solid areas, black deep holes, and white shallow holes. Overall, the Ti-Al-C-Nb samples prepared via ISLF had a porous structure inside, and their pores were a combination of small-area deep pores and large-area shallow pores. The pores on the sample surface with different process parameters changed regularly. The samples with a high scanning speed and a low laser power had more white areas, representing shallow holes. When the scanning speed was reduced and the laser power was increased, the white areas decreased, and the number and area of the shallow holes decreased.

Figure 10 shows that the Ti-Al-C-Nb samples processed with scanning speeds of 150 and 200 mm/s and laser powers of 40 and 45 W (the red framed area in Figure 10) had low shallow hole areas; extremely high and low laser energy inputs increased the areas of the shallow holes inside the samples, and affected the densities of the samples. Therefore, the correspondence between the laser energy density and the internal holes of the samples was considered. ImageJ software was used to extract the hole areas in all of the low-magnification SEM images in Figure 10.

Figure 11 presents the relationship between the average areas of the pores and the laser energy densities in the low-magnification SEM images (Figure 10). ImageJ software was used to extract the hole area in all of the low-magnification SEM images in Figure 10. One image per sample was used. Table 4 shows the data that were used to calculate the average areas of the holes. It can be seen from the figure that the sample with a laser power of 40 W and a scanning speed of 100 mm/s has the lowest average pore area. When the laser energy density was between 50 J/mm^2^ and 160 J/mm^2^, there was a decreasing trend between the average area of the pores and the laser energy density. The higher the laser energy density, the longer the existence time for the molten pool. The pores could be filled with the liquid phase, and the average area of the pores decreased. It should be noted that, when the laser energy density was between 160 J/mm^2^ and 200 J/mm^2^, the average area of the pores increased. This may have been caused by a small number of samples.

## 4. Discussion

### 4.1. Phase Diversification and TiC Change Law in ISLF

The changes in the XRD diffraction peaks in Section 3.1 indicated that chemical reactions occurred during ISLF. According to the results of the material system and the phase composition analysis, the possible reactions are as follows Equations (5)–(7) [20]:(5)$$\mathrm{Ti}+C=\mathrm{TiC}\;\Delta G_{1373K}^{0}=-168.190\mathrm{KJ}/\mathrm{mol}$$
(6)$$3\mathrm{Ti}+\mathrm{Al}=Ti_{3}\mathrm{Al}\;\Delta G_{973K}^{0}=-105.73\mathrm{KJ}/\mathrm{mol}$$
(7)$$\mathrm{Ti}+\mathrm{Al}=\mathrm{TiAl}\;\Delta G_{1000K}^{0}=-62.763\mathrm{KJ}/\mathrm{mol}$$

As an important parameter for measuring the spontaneous progress of the reaction, the smaller the thermodynamic Gibbs free energy (∆*G*), the easier the progression of the reaction. Reactions Equations (5)–(7) all have very small ∆*G* values. From a thermodynamic point of view, the three reactions had a strong tendency to proceed spontaneously, and Reaction Equation (5) preferentially proceeded with Reaction Equation (6), due to the small free energy value. In addition, all of the XRD patterns in Figure 4 were similar. An analysis suggested that the chemical reactions in ISLF were mainly Reactions Equations (5)–(7). The changes in laser power and scanning speed will affect the progress of the reaction, and thus affect the product content.

For the formation of the TiC phase, related studies [21] have shown that the reaction between Ti and C to synthesize TiC in the molten state has the lowest free energy, and that TiC is the most stable compound, and the easiest to form. The existence of a small amount of graphite phase in the XRD results of the ISLF sample indicates that the in situ synthesis reaction of TiC was not fully conducted. This analysis suggests two reasons. First, the temperature of the molten pool was probably insufficient, due to insufficient input energy, and the high-melt point graphite was not completely melted. Meanwhile, due to the poor wettability of the graphite and the molten aluminum [22], the low-melt point (660 °C) liquid Al that coated the Ti restricted contact between Ti and the graphite, and hindered the formation of the TiC phase. The scanning speed in Table 2 was constant. When the laser power increased from 35 W to 50 W, the graphite content decreased, and the corresponding increase in TiC content indicated that an increase in the energy input would promote the full progress of Reaction Equation (5).

In addition, the analysis of the TiC grains synthesized using different process parameters could better elucidate the in situ chemical synthesis process. The TiC grains underwent nucleation to form the growth–solidification process. During the rapid laser scanning process, the molten pool had a huge temperature gradient, and the TiC grains exhibited solidification–growth competition. Studies have shown that a working temperature under a low laser power limits the three-dimensional growth of TiC grains, and that TiC grains prefer two-dimensional layered growth [23]. Under the action of the low laser energy density of 60 J/mm^3^ within a short time, the temperature gradient of the molten pool changed greatly. After the crystal grains nucleated and solidified within a very short time, the growth time was very short, and so the TiC crystal grains appeared to be flat. The laser energy density *η* increased to 80 J/mm^3^, and the TiC grains grew in a three-dimensional space in spherical growth mode. Under the action of the recoil pressure generated inside the molten pool, and under the surface tension of the molten pool, the crystal grains in the matrix were evenly distributed. When the laser energy density increased further to 120 J/mm^3^, the temperature and the dynamic viscosity of the molten pool increased. Under the action of Marangoni convection in the pool, the spherical TiC grains were induced to form an orderly arrangement, forming multiple nanometers. The TiC crystal grains also had nascent nanocrystals that grew into microcrystals. When the laser energy density increased to 200 J/mm^3^, the temperature increased further, the internal stability of the molten pool decreased, and new nanocrystals were deposited onto the TiC strip crystals. The resulting temperature gradient led to the growth of the secondary crystal axis and finally formed thick dendrites.

As for the synthesis of the Ti-Al compound, the liquid Al that melted at nearly 700 °C warped the Ti powder. The Al/Ti ratio was the highest, and there was sufficient Al and Ti contact for a reaction. At this point in time, Reaction Equation (7) took precedence over Reaction Equation (6). However, as the temperature continued to rise, Reaction Equation (7) consumed part of the Al phase and volatilized the Al, and the Al/Ti content ratio decreased. Reaction Equation (6) was more competitive than Reaction Equation (7). At this time, Reaction Equation (6) between Ti and Al was dominant in the competition between Reaction Equation (6) and Reaction Equation (7). Unlike the traditional sintering method with slow heating, the ISLF process has a very rapid heating rate, and so Reaction Equation (6) took less time than Reaction Equation (6). During the laser scanning process, Reaction Equation (6) was the dominant reaction between Ti and Al, and it was the reason for why Ti_3_Al was the main phase in the ISLF samples and why TiAl was the secondary phase.

### 4.2. Formation of a Porous Structure

Figure 6 and Figure 7 show the surface morphologies of the ISLF samples. In the rough surface state, no obvious laser scanning path could be observed; that is, the melt channel. The V-shaped wheat ear shape was a typical melt channel morphology in the SLM process [24]. Although the SLM and ISLF processes were similar, the melt channel of the ISLF sample in this study did not show a continuous V-shaped morphology. An analysis revealed two reasons for such a structure. On one hand, the in situ chemical reaction that releases heat in the ISLF process increases the uncertainty of the melt channel morphology. On the other hand, the dynamic viscosity of the molten pool is also an important parameter that affects the competitive relationship between the spreading and the solidification of the liquid phase in the molten pool, and thus, it affects the quality of the sample.

From the perspective of the chemical reaction, the porous structure was mainly initiated from two aspects: the diffusion mechanism between the elements in the reaction process and the effect of the reaction on the molten pool. Figure 12 shows the SEM image of the ISLF sample surface and the EDS image of the corresponding area, with a laser power of 40 W and a scanning speed of 100 mm/s. The surface distribution results of the elements Ti, Al, C, and Nb in the EDS diagram showed that the TiC and Ti_3_Al synthesized by the chemical reaction of the sample were uniformly distributed during the ISLF process. Fine irregular particles could be observed on the surface of the sample in the SEM image, combined with the distribution of C, in Figure 12b,c. The results show that these particles are graphite powder. Considering the low density of graphite, graphite powder was inside the laser-induced molten pool. Splash occurred due to melt convection and thermal shock. Figure 12b shows that the Ti powder retained part of its original state and was half-hidden under the spheroidizing melt, indicating that the Ti powder did not melt completely within the very short laser action time. In our previous work [17], we mentioned that the diffusion of Al in Ti leaves many vacancies for forming pore defects, which are Kirkendall pores, due to the huge diffusion rate difference between Ti and Al. Meanwhile, the results of phase analysis indicated that the chemical reaction that occurred during the ISLF sample forming process, especially during the synthesis of TiC, released a large amount of heat, which diffused in the form of gas, increasing the instability of the molten pool and producing severe splashing, resulting in poor melt channel continuity. The Ti powder, which retained its original morphology, had insufficient energy to melt and to react with the original powder within a very short laser action time, approximately 10^−3^ s [25], proving that the temperature of the molten pool was insufficient.

The surface morphology results in Section 3.2 show that the surface of the ISLF sample had a porous structure with obvious spheroidization, with some irregular particles attached to the surface. In the process of laser selective formation, the dynamic viscosity of the molten pool directly affected the densification of the sample. According to the results of Takamichi and Roderick et al. [26], the dynamic viscosity (*μ*) inside the molten pool can be defined as follows:(8)$$\mu =a\sqrt{\frac{m}{kT}}\gamma ,$$
where *a* is a constant, *m* is the atomic mass, *k* is Boltzmann’s constant, *T* is the temperature of the molten pool, and *γ* is the surface tension of the liquid.

The formula indicates that molten pool temperature *T* and the liquid surface tension *γ* are two key factors that affect the dynamic viscosity *μ*. Increasing the scanning speed and reducing the laser power reduces the input laser energy and decreases the temperature of the molten pool accordingly, and the deterioration of the molten state of the powder causes the surface tension of the liquid to increase. Thus, the dynamic viscosity inside the molten pool increases. A high dynamic viscosity (*μ*) means that the liquid phase needs more time to spread. However, the large temperature gradient generated by the extremely short laser action time in the ISLF process will cause the molten pool to solidify rapidly. Generally, a competitive relationship exists between the spread of the liquid phase and the rapid cooling behavior in the molten pool [27]. In this study, the spreading time for the obvious droplets was significantly longer than their solidification time, indicating that the liquid phase had not completely spread before the solidification trend occurred, resulting in obvious spheroidization. Therefore, the results in Figure 6 and Figure 7 indicate that the laser energy increased from 35 W to 45 W when the scanning speed remained at 150 mm/s, and that the laser power remained unchanged at 45 W when the scanning speed decreased from 300 mm/s to 100 mm/s to increase the laser energy input. The temperature of the molten pool increased, the dynamic viscosity of the molten pool decreased, the spreading time of the liquid phase shortened, the spheroidization phenomenon was suppressed, the melting state of the powder layer improved, and the connectivity of the dispersed solidified melt was enhanced, but the powder layer was completely solidified. The morphology of the porous structure remained afterward. The next layer of powder melted to form a melt flow, and it filled the upper layer to form irregular holes. However, because the energy per unit area was limited, the highly inconsistent powder layer formed a molten pool with different dynamic viscosities under the action of the laser, and the powder did not melt completely. The degree worsened, and so the porous morphology of each layer accumulated after N times to form the surface morphology shown in Figure 6 and Figure 7.

From the above analysis, the reason for the porous structure of the sample can be obtained. The schematic of the formation of the porous structure of the ISLF sample is shown in Figure 13. First, the mixed powder, including the multiple phases, melted under the action of the laser. The large dynamic viscosity of the molten pool caused the droplet to solidify more strongly than the spreading behavior in the spreading–solidification competition behavior. The original powder could not be completely melted, and it began to re-solidify. Meanwhile, the chemical reaction between the original powders released a huge amount of heat, and the formed pores and Kirkendall pores led to the poor continuity of the molten pool after solidification, forming a melt channel with poor continuity and inconsistent width. Then, a single layer of powder formed via reciprocating laser scanning. Many irregular pores existed between the tracks (Figure 13b), and so the powder layer exhibited a different surface morphology after laser scanning. When the next layer of powder spread onto the spatula, although the upper surface of the new powder layer was flat, its internal height was not the same. The degree of laser melting for the different powder layers was obviously inconsistent. Although the laser scanning direction was rotated by 90°, this phenomenon greatly improved, but it was affected by the discontinuity of the melt channel itself and the accumulation of irregular holes between the melt channels in multiple layers after aggravating the appearance of the internal holes in the sample (Figure 13c).

## 5. Conclusions

In this study, the Ti-Al-C-Nb composite material was successfully prepared using the ISLF process, with mixed Ti, Al, graphite, and Nb powders as raw materials. The following conclusions are drawn:(1)The ISLF sample phase includes two main phases (i.e., Ti3Al and TiC): TiAl and the incompletely reacted graphite. The quantity of TiC is approximately 30–40%.(2)The microstructure of the in situ synthesized TiC phase changes with an increase in the laser energy density. At a low laser energy density, the TiC grains are mainly flat and spheroidal nano-level crystal grains, with a size as small as 152 nm. After increasing the laser energy density, the TiC grows into micron-level strip crystals and dendrites, with a size that can reach 5.01–7.05 μm.(3)The prepared ISLF sample has a porous structure, and the total open porosity is approximately 20% to 44%. The chemical reaction between the powders and the spheroidization caused by the high dynamic viscosity of the molten pool caused the powder layer to have an uneven, porous structure. The ISLF sample obtained after the accumulation of multiple layers has a porous structure.

## Figures and Tables

**Figure 1 materials-15-04351-f001:**
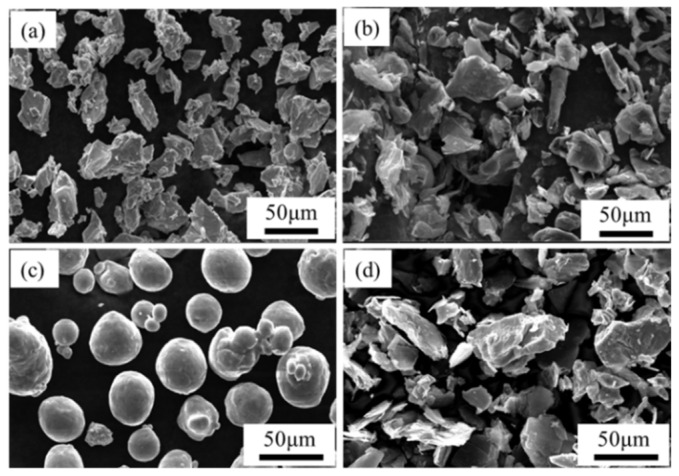
Morphology of raw materials: (**a**) Ti powder, (**b**) graphite powder, (**c**) Al powder, (**d**) Nb powder.

**Figure 2 materials-15-04351-f002:**
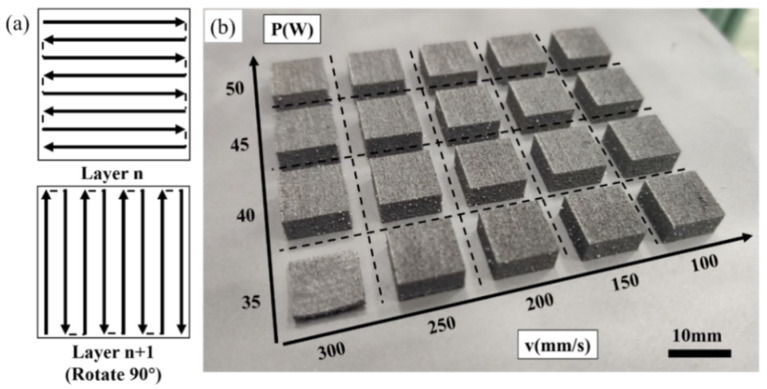
Laser scanning strategy and ISLF specimens: (**a**) “Zigzag” scanning strategy rotated by 90°; (**b**) ISLF samples.

**Figure 3 materials-15-04351-f003:**
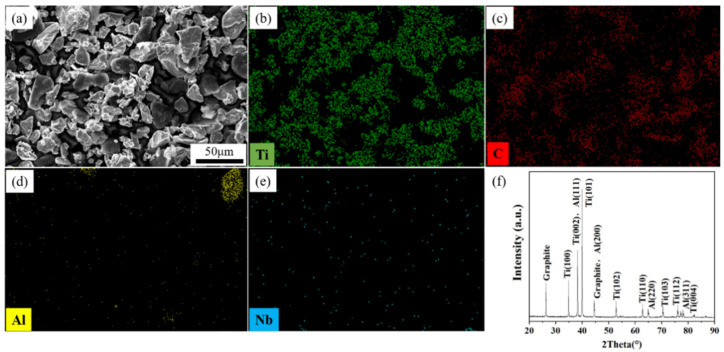
Mixed powder: (**a**) SEM image, (**b**) Ti element in EDS map, (**c**) C element in EDS map, (**d**) Al element in EDS map, (**e**) Nb element in EDS map, (**f**) XRD pattern.

**Figure 4 materials-15-04351-f004:**
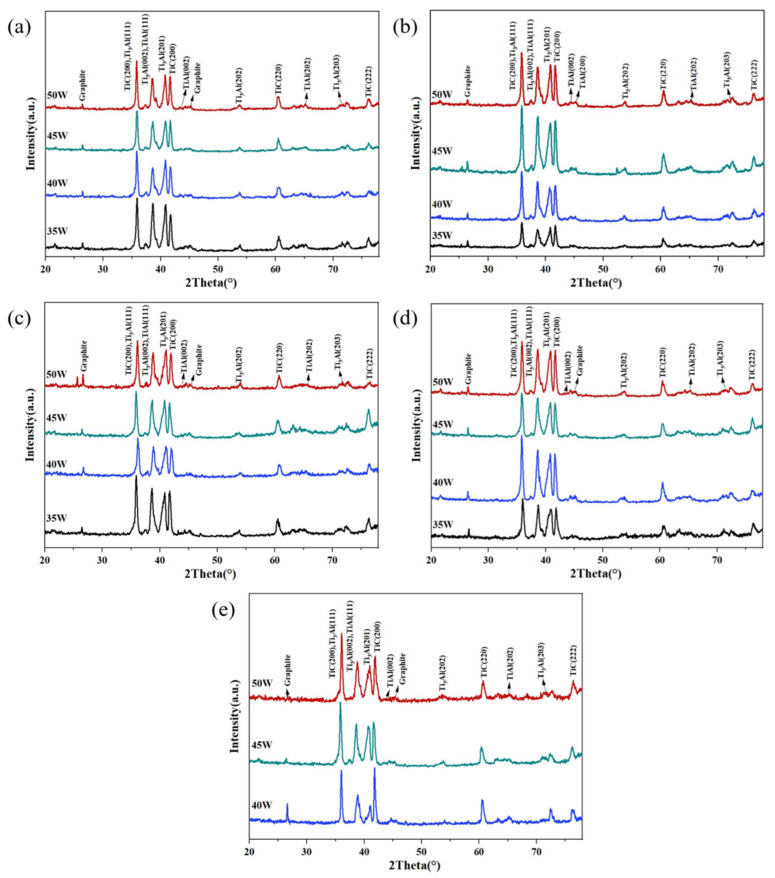
XRD results of ISLF samples with different process parameters: (**a**) 100 mm/s, (**b**) 150 mm/s, (**c**) 200 mm/s, (**d**) 250 mm/s, (**e**) 300 mm/s.

**Figure 5 materials-15-04351-f005:**
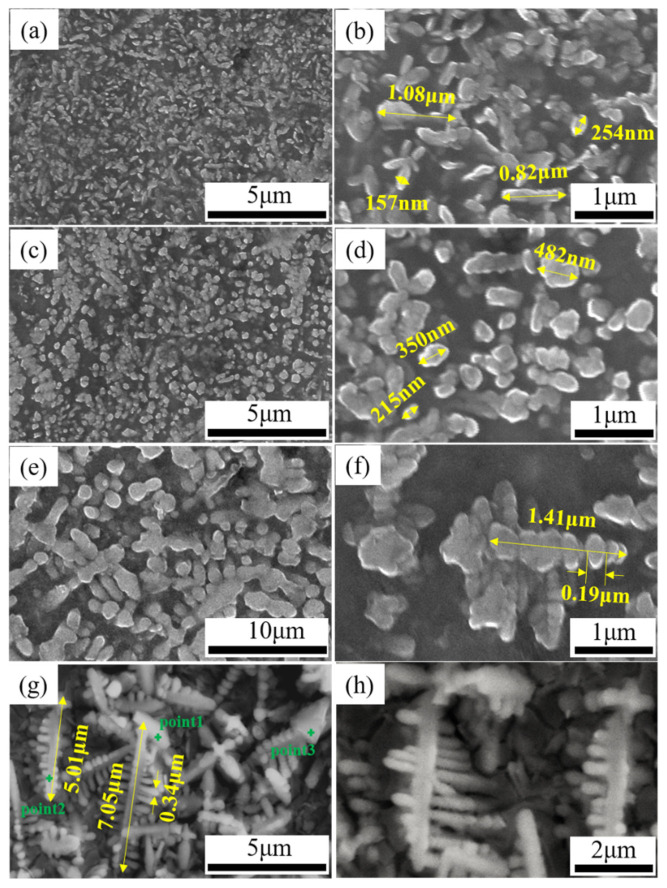
Microscopic morphology of TiC grains with different laser energy densities. (**a**,**b**) *η* = 60 J/mm^3^, (**c**,**d**) *η* = 80 J/mm^3^, (**e**,**f**) *η* = 160 J/mm^3^, (**g**,**h**) *η* = 200 J/mm^3^.

**Figure 6 materials-15-04351-f006:**
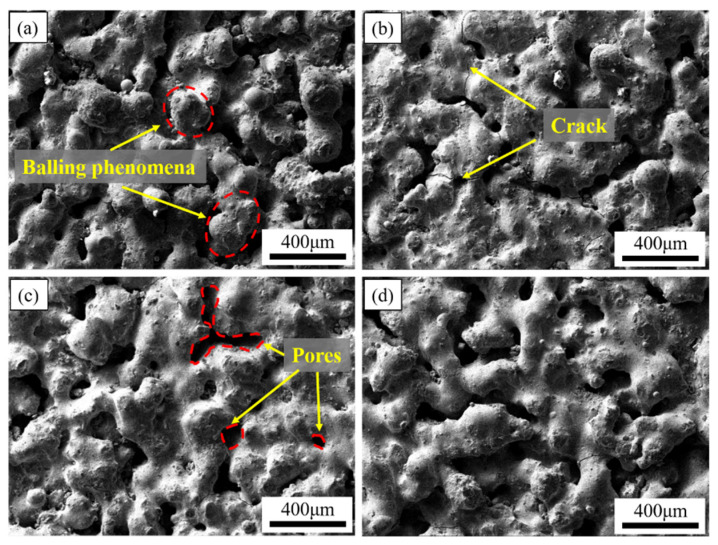
SEM images of the surface morphologies of the ISLF samples with different laser powers at a scanning speed of 150 mm/s: (**a**) 35 W, (**b**) 40 W, (**c**) 45 W, (**d**) 50 W.

**Figure 7 materials-15-04351-f007:**
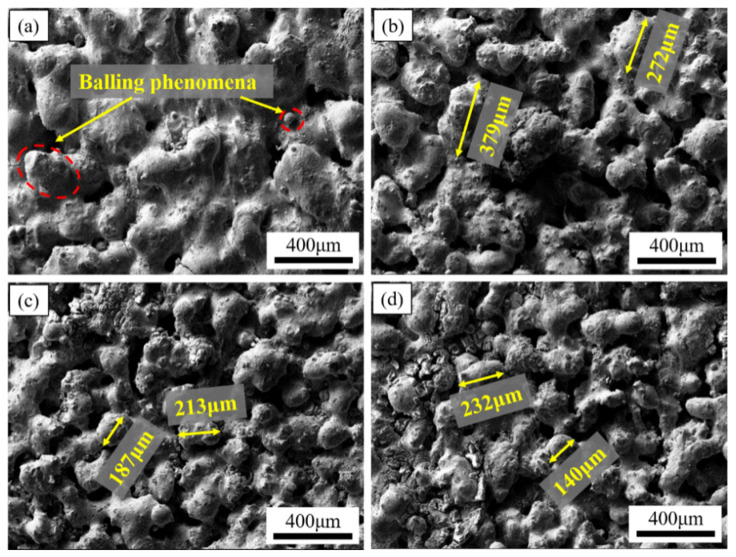
SEM images of Ti-Al-C-Nb sample surface morphologies at different scanning speeds with a laser power of 40 W: (**a**) 100 mm/s, (**b**) 200 mm/s, (**c**) 250 mm/s, (**d**) 300 mm/s.

**Figure 8 materials-15-04351-f008:**
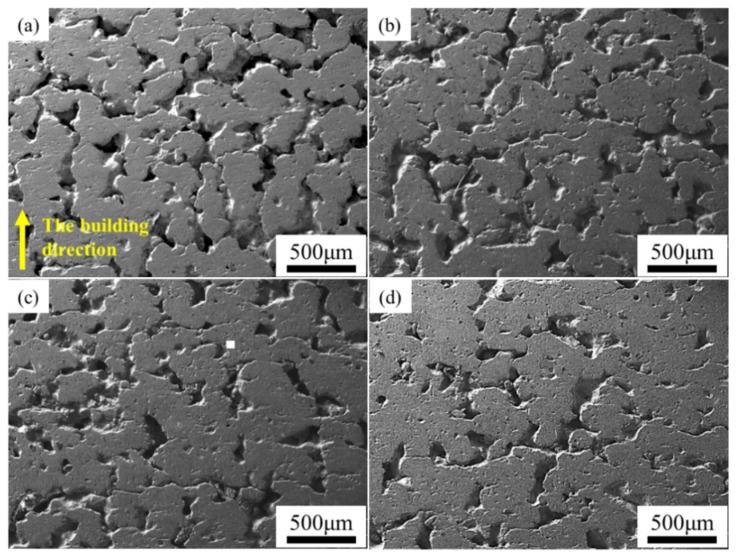
SEM images of the cross-sections of the samples. The scanning speed was 150 mm/s; the laser power values were (**a**) 35 W, (**b**) 40 W, (**c**) 45 W, (**d**) 50 W.

**Figure 9 materials-15-04351-f009:**
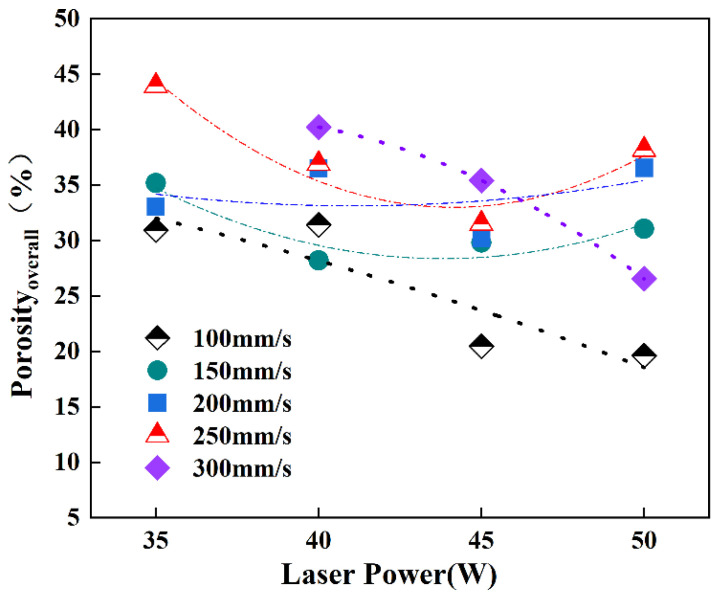
Overall porosities of ISLF samples with different processing parameters.

**Figure 10 materials-15-04351-f010:**
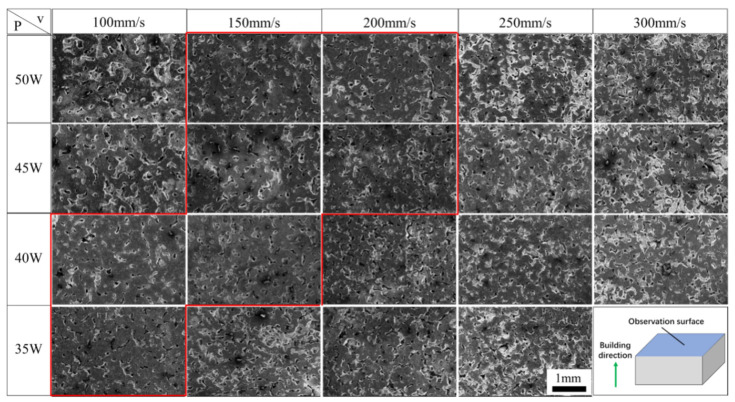
Low-magnification SEM images of ISLF samples.

**Figure 11 materials-15-04351-f011:**
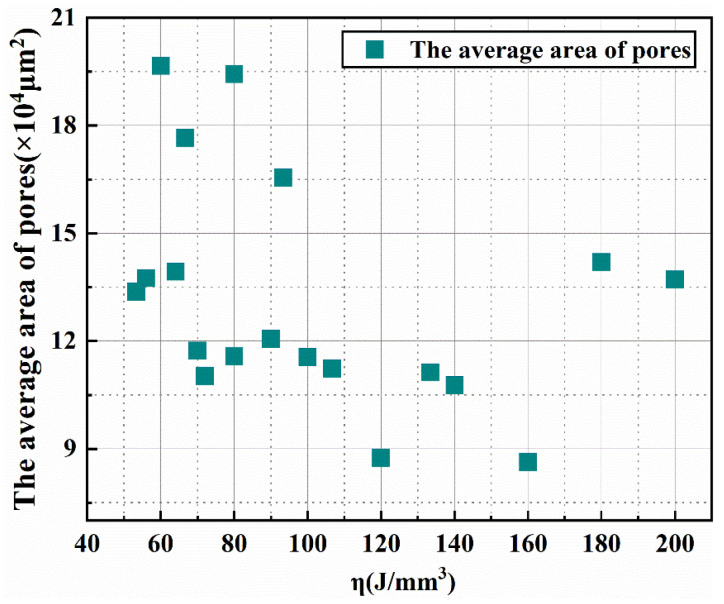
Average area distributions of the pores in the low-magnification SEM images of the ISLF samples.

**Figure 12 materials-15-04351-f012:**
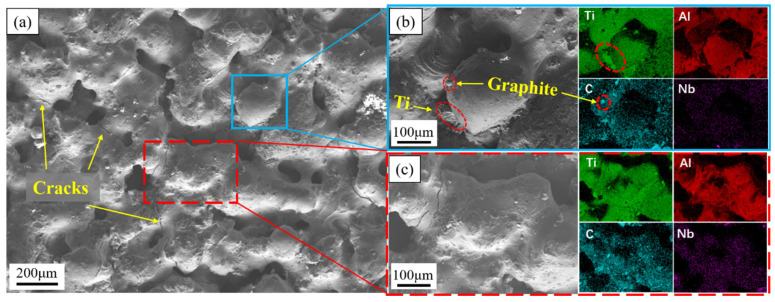
ISLF sample surface of 40 W 100 mm/s: (**a**) SEM image, (**b**) partially enlarged image and EDS image, (**c**) partially enlarged image and EDS image.

**Figure 13 materials-15-04351-f013:**
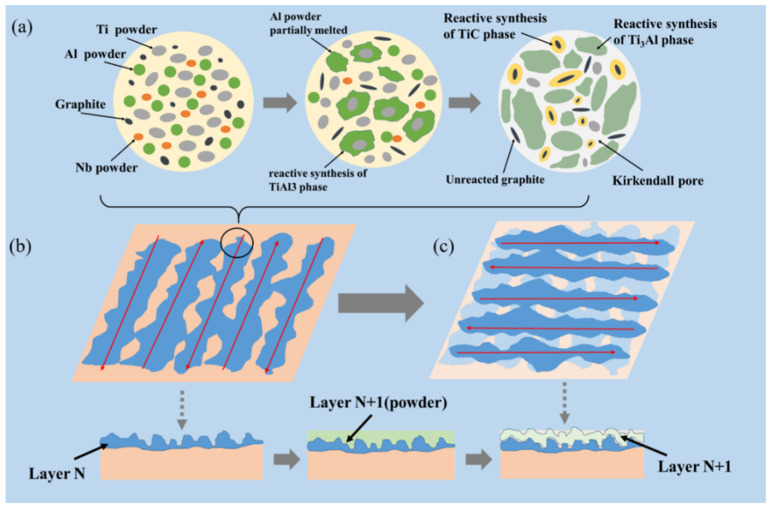
Schematic of the porous structure formation of ISLF sample. (**a**) ISLF reaction process; (**b**) Melt channel of layer N; (**c**) Melt channel of layer N+1.

**Table 1 materials-15-04351-t001:** Laser energy density *η,* corresponding to different process parameters.

Laser power	50 W	50 W	50 W	50 W	50 W
Scanning speed	100 mm/s	150 mm/s	200 mm/s	250 mm/s	300 mm/s
*η* (J/mm^3^)	200	133.33	100	80	66.67
Laser power	45 W	45 W	45 W	45 W	45 W
Scanning speed	100 mm/s	150 mm/s	200 mm/s	250 mm/s	300 mm/s
*η* (J/mm^3^)	180	120	90	72	60
Laser power	40 W	40 W	40 W	40 W	40 W
Scanning speed	100 mm/s	150 mm/s	200 mm/s	100 mm/s	300 mm/s
*η* (J/mm^3^)	160	106.67	80	64	53.33
Laser power	35 W	35 W	35 W	35 W	35 W
Scanning speed	100 mm/s	150 mm/s	200 mm/s	250 mm/s	300 mm/s
*η* (J/mm^3^)	140	93.33	70	56	46.67

**Table 2 materials-15-04351-t002:** Phases quantities of each phase in the ISLF sample, corresponding to different process parameters.

*P* (W)	*v* (mm/s)	Phases Quantities, wt.%				Rwp	Rexp	χ^2^
		Ti_3_Al	TiC	TiAl	Graphite			
35	100	38.81	31.74	22.94	6.51	22.34	19.35	1.33
40	100	52.30	31.40	12.00	4.28	25.46	21.79	1.37
45	100	36.40	34.82	22.27	6.49	25.00	22.90	1.19
50	100	43.10	36.70	14.50	5.70	22.90	20.87	1.20
35	150	40.50	33.69	16.26	9.51	26.37	23.36	1.27
40	150	38.36	32.60	22.33	6.70	19.12	15.69	1.49
45	150	39.40	32.28	21.90	6.41	21.09	19.10	1.22
50	150	37.86	35.04	24.70	2.40	21.82	14.83	1.40
35	200	42.40	29.99	21.94	5.67	22.96	19.00	1.46
40	200	44.20	34.60	13.00	8.20	28.35	26.54	1.14
45	200	40.60	32.33	22.22	4.85	20.69	18.4	1.26
50	200	44.30	30.20	14.70	10.80	33.02	26.87	1.51
35	250	49.30	29.82	13.12	7.80	27.40	25.10	1.19
40	250	41.95	32.67	19.60	5.78	20.87	17.89	1.36
45	250	38.78	31.08	23.91	6.23	21.73	18.56	1.37
50	250	50.00	31.62	16.62	1.77	24.44	18.97	1.66
40	300	20.40	40.27	20.10	19.30	27.84	27.17	1.05
45	300	43.96	34.15	16.38	5.52	22.21	19.08	1.36
50	300	44.89	38.98	14.26	1.87	26.29	22.93	1.31

**Table 3 materials-15-04351-t003:** Elemental contents for different points obtained by EDS in Figure 5g.

Point	Atomic Percent, %		
	C	Ti	Al
Point1	45.0	46.2	8.8
Point2	47.0	42.8	11.2
Point3	36.4	46.9	16.7

**Table 4 materials-15-04351-t004:** The data used to calculate the average area of the holes.

Laser Power (W)	Scanning Speed (mm/s)	Pixels (px)		Pixels per Unit Length mm (px/mm)	Porosity (%)	Total Area of Holes (mm^2^)	Number of Holes	Average Area of the Holes (μm^2^)
		*b*	*c*	*d*	*p*		*n*	*S*
35	100	1534	1022	371	37.751	4.3076149	40	107,690.37
40	100	1536	1025	371	34.654	3.9710066	46	86,326.23
45	100	1534	1023	371	41.003	4.6832655	33	141,917.14
50	100	1535	1026	371	37.071	4.249347	31	137,075.71
35	150	1535	1023	371	50.672	5.7914087	35	165,468.82
40	150	1535	1025	371	42.16	4.8279749	43	112,278.49
45	150	1534	1020	371	24.566	2.797642	32	87,426.311
50	150	1535	1024	371	39.886	4.5631104	41	111,295.38
35	200	1534	1025	371	36.898	4.2226414	36	117,295.6
40	200	1536	1022	371	40.499	4.6272031	40	115,680.08
45	200	1533	1023	371	42.264	4.8241471	40	120,603.68
50	200	1531	1024	371	40.507	4.6220792	40	115,551.98
35	250	1536	1022	371	46.935	5.3625467	39	137,501.2
40	250	1535	1024	371	43.841	5.0155775	36	139,321.6
45	250	1534	1023	371	36.68	4.1895027	38	110,250.07
50	250	1535	1020	371	47.747	5.4411011	28	194,325.04
40	300	1536	1018	371	41.122	4.6799949	35	133,714.14
45	300	1535	1026	371	48.037	5.5063495	28	196,655.34
50	300	1534	1022	371	46.412	5.2958868	30	176,529.56

## Data Availability

Not applicable.

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
