# Peer review of "Microstructure and Phase Evolution of Ti-Al-C-Nb Composites Prepared by In Situ Selective Laser Forming"

_materials, 2022, doi:10.3390/ma15124351_

Round 1

Reviewer 1 Report

Thank you very much for submitting paper. Author showed about SEM image and XRD of the ISLF specimen. But author should be consider about following comments.

1.     Author must be changed about title.

2.     Fabrication’s information is low in this paper.

3.     Author described “The TiC crystal grains show different distributions of microscopic morphology with the increase of the laser energy density” (p7). The layer in microstructure of sample is not clear. Author should be showed about values of thickness of some layers in microstructure. And the scatterings of layers at some areas is not clear.

4.     Author showed some images of surfaces of samples in figure 7. Author described “The figure shows that the surface of the ISLF sample is rough and porous. Obvious spheroidization and many irregular pores can be observed on the surface of the sample.” But many data of pores and creak in composites are no clear.

5.     Author showed about porosity of ISLF sample opening rate with different process parameters in figure 8. The mechanism on porosity of ISLF sample were not clear when range of laser power were 35 W to 50 W. Author must be described the mechanism of overall porosity of ISLF sample under laser powers.

6.     Author must show the statistical data (shape, grain size, strip crystals, dendrite length and so on) of sample. (Figure 5)

7.     Author must be showed about standard division of molar fraction of phases of sample.

8.     Figures 6(b) and 7(b) are same photo.

9.     Author must be described about definition of hole area.

10.  Average area distribution of the pores in the low-magnification SEM of the ISLF sample is shown in figure 10. Curve is not good. Author must be described about mechanism of area distribution of the pores of the ISLF sample under 160-200J/mm2

11.  Author described “original powder within a very short laser action time” (p12). author must be described about value of very short laser action time.

Reviewer 2 Report

The work deals with the investigation of Ti-Al-C-Nb composites prepared by in-situ selective laser forming. The main aim was to study the influence of process parameters on the phase composition and microstructure of the prepared samples. The manuscript is well written and organized, the results are described in detail and clearly discussed. However, some major revision is mandatory. The main comments are given below:

  1. The results of the sample prepared at 35W laser power and 300 mm/s scanning speed are omitted in the manuscript. Since the sample is listed among the other samples, the results should be included in the manuscript.
  2. XRD patterns comprise the phase assignment, however, the Miller indices are missing. To increase the scientific value, the particular diffracting planes should be given to the each phase. Moreover, colors of the XRD patterns should be consistent (Fig. 4e).
  3. Table 1 should be re-organized. I would suggest to start new line with different laser power. This will make the table visually clearer. Moreover, each set of parameters (including laser power, scanning speed and energy density) should be separated with a solid line. The “Parameter” in the table should be specified for the each line.
  4. Authors stated: “White particles are mainly TiC grains” (lines 171-172). What is the experimental prove of this statement? According to Table 2, the Ti3Al phase has higher molar fraction compared to TiC. Therefore, I would expect the dominant occurrence of Ti3Al in the microstructures. Besides that, TiAl3, graphite and Nb should occur.
  5. Authors should specify, how they determined molar fractions of particular phases given in Table 2. If the ImageJ software was used, how many images per sample were used? In that case, the standard deviation should be included.
  6. Lines 212-213: The trend potentially showing the decrease in overall porosity with the increase of laser power is not unambiguous. This trend is valid only for the 300 mm/s scanning speed results, however, the value at 35W is missing. Please, give a statement. Moreover, the interconnection of measured points is not relevant as the probability of a linear course between two points is very low. If authors want to show the trend, an approximation function should take place.
  7. How many images per sample were measured to extract the holes area? Due to the possible heterogeneity of the sample, the mean values including their standard deviations should be considered.
  8. The discussion focuses mainly on the TiC phase and there is no mention of Nb. Authors should add the brief discussion on the distribution of Nb and its potential influence on the formation of microstructure, for example in comparison with Ti-Al-C composites studied previously. Subsequently, it should be reflected in conclusion (1).
  9. The manuscript also needs some minor formal editing:

- text formatting should include superscripts and subscripts, space below figure captions, the same figure captions font as used in the text,

- quantities of formula given in the text should be written in italics,

- the stoichiometry in the case of the “Ti-Al” phase (line 27) is not clear. This form of designation comprising “-“ is more reminiscent of an alloy system rather than a specific phase. Please, specify.

- line 72: the symbol “-“ used in front of the particle size value is confusing. If it means that the particle size is lower than given values, authors could use the symbol “<” instead,

- lines 103, 188-189: “samples” instead of “sample”,

- line 119: “Cu target”, better: “Cu anode”,

- lines 128-129: “energy spectrum detector system”, better: “energy-dispersive X-ray detector”,

- line 130: “ImageJ” instead of “Image-J”,

- line 131: it is better to use “etch” instead of “corrode”,

- lines 145-146: it is better to use “EDS map” and “XRD pattern”,

- Table 2: V ---> v,

- line 255: please, fill the brackets,

- line 262: “related studies show [20]”, better: “related studies [20] show”.

Reviewer 3 Report

The authors have studied a novel Ti-Al-C-Nb composite prepared through in-situ selective laser forming. The microstructure, surface morphology, EDS, and XRD analyses, as well as porosity test were performed and interesting results were obtained. The work is well done, but some deficiencies need to be corrected to make the manuscript acceptable for publication.

(1) Line 96: The corrections should be made as follows: “where ɳ represents the energy density (J/mm3),  …, and h represents the layer thickness (mm)”.

(2) Formula (2): Please check the spelling, namely, “ethanal” should be replaced with “ethanol”.

(3) Please check the sequence of the letters (a), (b), (c), (d), (e) in the caption to Fig. 7.

(4) The caption to Fig. 8 is confusing.

(5) Lines 52–55: It is expected that among References provided in this manuscript, to gain more recent references in the field, the following reference be added: https://doi.org/10.1016/j.compstruct.2021.114649

It may be as follows: “Although the TiAl phase, which is one of the multiple phases in the Ti-Al-C system [the above reference], …”

Round 2

Reviewer 1 Report

Thank you very much for some comments. The data of the average area of the holes in all low-magnification SEM images to plot the line chart is important for paper.

1.The mechanical properties of ISLF sample is affected by hole shape. Creak length and holes shape of sample at thickness direction are not cleared for mechanism.

2.Please add about table and method(figure) of holes in all low magnification SEM images for paper or supplemental data (figures 3-4 and 3-5).

3.Author described about ranges of some values. But, their standard divisions in statistical data are important for probability. Author must be showed the value of standard division of data.

Reviewer 2 Report

The authors reflected most of the comments in the revised manuscript. However, some more revision is requested before the manuscript can be published. The comments are given below:

1. Please avoid symbols in the XRD patterns and use the direct phase denotation in front of the Miller indices (Fig. 4). The same approach should be reflected in Fig. 3f.

2. Table 1: Instead of “Parameter” through two lines, authors could use “Laser power” and “Scanning speed” separately for the each line.

3. Although authors gave the relevant explanation for the presence of TiC (as white particles) in Fig. 5, it is still only an assumption without experimental approach. The EDS maps (similar to Fig. 3 or Fig. 11) at least for one representative sample would clarify the distribution of elements. Thus, taking into the account the EDS results combined with XRD, the microstructure (white particles) could be properly described. 

4. Measured points in Fig. 8 should not be interconnected with solid lines as it is not scientifically correct. The points should stay separated, since the probability of a linear course between each two interconnected points is very low. If authors want to highlight the trend, they should use an approximation function instead.

5. Please, write also the quantities in the text in italics (e.g. lines 103-105).

Reviewer 3 Report

(1) In the added reference [26], https://doi.org/10.1016/j.compstruct.2021.114649, the authors’ names should be corrected: Please replace “Tap A , Opo B , Ask C , et al. ...” (incorrect names) with “Prikhna T A , Ostash O P , Kuprin A S , et al. A new MAX phases-based electroconductive coating for high-temperature oxidizing environment [J]. Composite Structures, 2021, 277: 114649.” (correct names).

The reviewer’s comments were taken into account. Now this paper may be published in the journal “Materials”.

Round 3

Reviewer 1 Report

Thank you very much for some comments.

Author Response

We sincerely thank the editor and the reviewers for their comments, which are very helpful in improving the quality of the manuscript. 

Point 1: (x) Moderate English changes required
Response 1: We thank the reviewer’s kindly suggested. This paper has undergone English language editing by MDPI these days. 

Reviewer 2 Report

Please, round the values of chemical composition in Fig. 5i to one decimal place, as it is the accuracy of the EDS method. Besides that, the (in)accuracy of detection of C by EDS should be taken into the account...

Consider putting the measured values in a separate table instead of part of the image.

After this small minor revision, the manuscript can be accepted for publication, since all my questions and comments have been properly answered.

Author Response

We sincerely thank the editor and the reviewers for their comments, which are very helpful in improving the quality of the manuscript.

Reviewer 3 Report

The reviewer’s comments were taken into account. Now this paper may be published in the journal “Materials”.

Author Response

(The authors gave the same response as above.)
